# Increased Risk of Sub-Clinical Blood Lead Levels in the 20-County Metro Atlanta, Georgia Area—A Laboratory Surveillance-Based Study

**DOI:** 10.3390/ijerph18105163

**Published:** 2021-05-13

**Authors:** Carmen M. Dickinson-Copeland, Lilly Cheng Immergluck, Maria Britez, Fengxia Yan, Ruijin Geng, Mike Edelson, Salathiel R. Kendrick-Allwood, Katarzyna Kordas

**Affiliations:** 1Department of Microbiology, Biochemistry, and Immunology, Morehouse School of Medicine, Atlanta, GA 30310, USA; limmergluck@msm.edu (L.C.I.); mbritez@msm.edu (M.B.); 2Department of Community Health and Preventive Medicine, Morehouse School of Medicine, Atlanta, GA 30310, USA; fyan@msm.edu; 3Pediatric Clinical and Translational Research Unit, Clinical Research Center, Morehouse School of Medicine, Atlanta, GA 30310, USA; rgeng@msm.edu; 4Geographic Information Systems, InterDev, Roswell, GA 30076, USA; medelson@interdev.com; 5Department of Pediatrics, Divisions of General Pediatrics & Neonatology, Emory University School of Medicine, Atlanta, GA 30303, USA; salathiel.r.kendrick-allwood@emory.edu; 6Department of Epidemiology and Environmental Health, University at Buffalo, Buffalo, NY 14214, USA; kkordas@buffalo.edu

**Keywords:** lead poisoning, childhood exposure, multilevel regression analysis, GIS

## Abstract

Lead (Pb) is a naturally occurring, highly toxic metal that has adverse effects on children across a range of exposure levels. Limited screening programs leave many children at risk for chronic low-level lead exposure and there is little understanding of what factors may be used to identify children at risk. We characterize the distribution of blood lead levels (BLLs) in children aged 0–72 months and their associations with sociodemographic and area-level variables. Data from the Georgia Department of Public Health’s Healthy Homes for Lead Prevention Program surveillance database was used to describe the distribution of BLLs in children living in the metro Atlanta area from 2010 to 2018. Residential addresses were geocoded, and “Hotspot” analyses were performed to determine if BLLs were spatially clustered. Multilevel regression models were used to identify factors associated with clinical BBLs (≥5 µg/dL) and sub-clinical BLLs (2 to <5 µg/dL). From 2010 to 2018, geographically defined hotspots for both clinical and sub-clinical BLLs diffused from the city-central area of Atlanta into suburban areas. Multilevel regression analysis revealed non-Medicaid insurance, the proportion of renters in a given geographical area, and proportion of individuals with a GED/high school diploma as predictors that distinguish children with BLLs 2 to <5 µg/dL from those with lower (<2 µg/dL) or higher (≥5 µg/dL) BLLs. Over half of the study children had BLLs between 2 and 5 µg/dL, a range that does not currently trigger public health measures but that could result in adverse developmental outcomes if ignored.

## 1. Introduction

Although significant progress has been made to reduce childhood exposure, lead (Pb) still poses a threat to children’s health and environmental justice [1]. Pb is a pervasive environmental contaminant and potent neurotoxicant in children that accumulates in bones over long periods of time [2,3,4,5]. Pb can impact child health across the entire range of possible exposures, and there is no safe level for children. Federal regulation has drastically reduced the prevalence of elevated blood lead levels (BLLs) in the U.S.; however, BLLs below the clinically relevant threshold still have a considerable impact on children [6,7]. BLLs < 10 µg/dL can result in nervous system damage, speech, language, and behavioral problems including attention deficit hyperactivity disorder, lower IQ, and learning disabilities [8,9,10]. In 2012, the reference value for elevated BLLs in children was lowered by the Centers for Disease Control and Prevention (CDC) from 10 µg/dL to the current actionable level of 5 µg/dL [11]. This value, however, does not reflect the adverse effects of low-level lead (LLL), documented at BLLs as low as 2 µg/dL or lower [12,13,14].

Current screening and testing guidelines recommend testing children at ages 12 and 24 months and taking action with a BLL of 5 µg/dL or higher [15]. These guidelines leave children, especially older children for whom no federal guidelines exist, at risk of chronic LLL exposure [16,17]. In addition to the screening gaps introduced by the current guidelines, three obstacles within the healthcare system exacerbate the gap. First, only 10 states, plus the District of Columbia, have universal pediatric lead testing policies [18]. Furthermore, states do not follow uniform practices for screening, testing, and reporting of BLLs [17,19]. Second, since testing is healthcare-provider-driven, and many children are not considered “high-risk” for elevated BLLs based on self-reported lead exposure risk questionnaire, they are never tested and their exposures are underreported [20,21]. Finally, decreased sensitivity of BLL tests at the lower levels of exposure when point-of-care technologies are employed means decreased reliability to detect levels below 3 µg/dL [22]. To effectively prevent adverse outcomes associated with lead exposure, the prevalence of and risk factors for exposures that fall below the current CDC actionable level must be accurately characterized.

In Georgia, and metro Atlanta specifically, the gentrification of older neighborhoods with Pb-laden soil, and housing dust increases the risk of chronic Pb exposure in children across socioeconomic classes. The societal cost in the form of lost productivity and human potential is extensive as chronic LLLs continue unchecked [23]. The objective of this study was to identify predictors for sub-clinical BLLs, between 2 and 5 µg/dL, as children with these BLLs are not currently considered “poisoned” but may still experience adverse effects. We used Georgia Department of Public Health Lead Prevention Program (GDPH HHLPP) data from 2010 to 2018 to describe the distribution of BLLs in children aged 0–72 months and examine sociodemographic and area-level characteristics associated with BLLs 2 to <5 µg/dL, considered sub-clinical based on current CDC guidelines, and BLLs ≥ 5 µg/dL, the current actionable level. Additionally, we performed geographic information systems (GIS) mapping to identify hotspots of sub-clinical (2 to <5 µg/dL) and actionable (≥5 µg/dL) BLLs in that population to investigate significant spatial relationships in exposure. Previous lead exposure studies have focused on elevated BLLs and used their associated risk factors to identify “high-risk” children [24,25]. However, such findings do not characterize children at risk for sub-clinical BLLs to drive timely intervention and prevention, thus leaving large groups at risk of harm.

## 2. Materials and Methods

### 2.1. Study Design

The dataset was obtained from the Georgia Department of Public Health’s Healthy Homes for Lead Prevention Program (GDPH HHLPPP) surveillance database. This dataset compiled BLLs from healthcare providers and independent laboratories that used self-reported data on child’s age in months, gender, address, self-reported race, and Medicaid status. A unique ID was assigned for each child, and the following variables: age, gender, race, residential home address, and collection year, were retrieved for each child. The raw dataset provided laboratory-based BLL results and demographic characteristics for 491,973 children aged 0–18 years old from the 20-county metro Atlanta area during the period from 2010 to 2018.

For this study, our exclusion criteria were: (1) individuals with BLLs from blood sources other than venous draw (*n* = 354,380), (2) children aged more than 72 months old or missing age (*n* = 11,378), (3) incomplete or missing address data (*n* = 5393), and (4) duplicates with multiple BLLs in a calendar year (*n* = 27,660). These exclusions yielded a study sample of 93,162 cases (Figure 1).

### 2.2. Study Population and Variables

Based on the current guidelines to screen children at 12- and 24-month well-child checks, the study population was divided into two age groups: 0–2 years (0–24 months), and 2–6 years (25–72 months) (Figure 1). The BLL was split into three categories based on the lower limit at which physiological and behavioral outcomes have been documented (2 µg/dL), and the current reference level for public health intervention (5 µg/dL). There is not a consistent lower level of detection among laboratories; therefore, the first category consists of BLLs < 2 µg/dL, the second category contains BLLs from 2 to <5 µg/dL (sub-clinical), and the third category are BLLs ≥ 5 µg/dL.

Sociodemographic, individual-level variables were categorized as: age (0–24 months, and 25–72 months), sex (male and female, or missing), race (re-categorized as Black, White, Others (including American Indian or Alaska Native, Asian or Pacific Islander, and Multiracial) and Unknown (those without response or missing information)), Medicaid status (yes/no, indicating program enrollment), and place of residence based on zip code (defined as Urban, Suburban, or Rural).

Area-level variables were obtained using the 2010 U.S. Census tract information at the block and block group level. We then overlaid the Census data on several other publicly available databases including the 2017 American Area Census Survey; Age less than 5 years (%) and Age 5–9 years (%), White (%), Black (%), No High school graduate (%), Total Housing Units; Esri Enrichment Income below the poverty level (%), GED or High School diploma (%), College degree (%), 2019 Total Crime Index (Calculated by ArcGIS, based on careful compilation and analysis of the FBI Uniform Crime Report databases. The index values for the U.S. level are 100, representing average crime for the country. A value of more than 100 represents higher crime than the national average, and a value of less than 100 represents lower crime than the national average), 2010 Homeowner (%), 2010 Renter (%), 2018 American Community Survey; Single-family home (%), Multi-unit homes (%), Household headed by a single female (%), and the 2010 U.S. Census tract (GIS and zip code information).

### 2.3. Spatial Analyses

The Optimized Hotspot Analysis (OHSA) tool (Esri ArcGIS Pro, Redlands, CA, USA) was used to visualize spatial patterns, such as clustering, of cases of sub-clinical and clinical BLLs. We utilized the Getis-Ord Gi* hotspot analysis statistic, inputting a residential address and BLL to evaluate the statistically significant spatial clustering of high and low values. The analysis allowed us to visualize the locations within the twenty-county study area where hot- or coldspots occur within a “cluster” in a non-random way. The OHSA requires the users to provide at minimum three variables: (1) patient location (in the form of an address or coordinates), (2) a defined Study Area, and (3) a designated Neighborhood size.

For the ArcGIS Pro Geocoding Tool (Esri), we used a study population of 93,162 patients and selected the years 2010 and 2018. We categorized the age variable (0–24 months and 25–72 months) and the BLL outcome (sub-clinical BLL 2 to <5 µg/dL and clinical BLL ≥ 5 µg/dL) for comparison of trends. The study area included 20 Georgia counties, in or near the Atlanta Metropolitan Statistics area. Lastly, we defined a Neighborhood area as a 1.5-mile radius around the child’s residential location using a hexagon grid pattern. The 1.5-mile radius accounts for variations in the sizes of neighborhoods, apartment complexes, and other living quarters across the metro Atlanta area. When executed, the OHSA tool associates the patient with the neighborhood and then compares the neighborhood to the rest of the study area. If the number of cases in a neighborhood is higher than the rest of the study area, then the patient is in a hotspot. Conversely, if the number of cases is much lower in a neighborhood, then the patient is in a coldspot.

### 2.4. Data Analyses

For statistical analysis, BLL was split into three categories representing: (1) the lower limits of detection, BLLs < 2 µg/dL; (2) the current threshold for intervention, BLLs ≥ 5 µg/dL; and (3) the range of values between these thresholds; BLLs 2 to <5 µg/dL.

Descriptive statistics were performed for both individual-level and area variables for the three BLL categories (<2 µg/dL, 2 to <5 µg/dL, and ≥5 µg/dL). Mean with standard deviation or median with interquartile range (IQR) was used for continuous variables, and frequency with percentage was used for categorical variables.

To consider both individual and area-level variables as potential predictors of lead exposure, four hierarchical models were performed to compare sub-clinical BLLs versus BLLs < 2 µg/dL, and sub-clinical BLLs versus BLLs ≥ 5 µg/dL. For each comparison, Model 1 is the base model to show the variability in the likelihood of children from each blockID having a BLL 2 to <5 µg/dL. In Model 2, individual-level variables were added to the blockID to estimate the association between individual-level variables and the likelihood of having sub-clinical BLLs compared to either BLLs < 2 µg/dL or BLLs ≥ 5 µg/dL. In Model 3, area-level variables were included together with blockID. Model 4, the full model, included blockID and both individual and area-level variables. Proportional change in variance (PCV) was calculated to show the percentage of variation contributed from individual or area variables, and intra-cluster correlation (ICC) was calculated to describe the variability due to area-level factors. Log-likelihood and AIC value were obtained to describe the model fit. All analyses were performed using SAS 9.4 (SAS Institute Cary, NC, USA), and confidence interval was used to consider statistical significance.

## 3. Results

### 3.1. Spatial Analyses

The Geographic Information Systems Hotspot Analysis indicated changes in the geospatial distribution of BLLs in the 20-county metro Atlanta area from 2010 to 2018. The hotspots analyses depict geographic areas with clustering of BLLs with 90–99% confidence based on the cumulative incidences of sub-clinical (BLLs 2 to <5 µg/dL) or clinical BLLs (BLLs ≥ 5 µg/dL) for each block group in the 20-county metro Atlanta study area. The maps contained in Figure 2 and Figure 3 are subdivided by age and BLL categories to visualize the trends in the study population across this period.

The spatial distributions of sub-clinical and clinical BLLs indicate distinct patterns for the years 2010 and 2018. In both years, the numbers of sub-clinical cases were higher than clinical cases for children aged <2 and 2–6 years and therefore appeared as more dense clusters. However, the density of cases with BLL 2 to <5 µg/dL and ≥5 µg/dL appears to have reduced by 2018. The latter year also includes a coldspot in the southern rural region, indicating that sub-clinical cases among 2–6-year-olds occurred in this area, but at a lower frequency than elsewhere in the Atlanta metro area. For both categories of lead and particularly for children aged 2–6 years, the location of the cases became more diffuse by 2018, appearing in more counties and further away from central areas of Atlanta.

### 3.2. Participant Characteristics, by Lead Status

Our study population showed similar distribution across those who were less than 24 months and those who were 25–72 months, and proportionally higher rates of elevated lead levels in the older age group. The percentage of Georgia Department of Public Health’s Healthy Homes and Lead Prevention Program (GDPH HHLPPP) participants aged 0–24 months and 25–72 months with valid venous BLLs from the 20-county metro Atlanta area were 54.7% and 45.3%, respectively, over the 9 years from 2010 to 2018. Regardless of the BLL outcome, the participants in each sub-category of BLL were younger and had roughly the same number of boys (51.2–52.3%) and girls. Although Blacks comprised 34.6% of the study sample, they were more frequently observed in the BLL < 2 µg/dL group (49.9%) compared to other racial/ethnic groups. Children with Medicaid-sponsored healthcare coverage accounted for 73.7% of the study population. Most participants resided in suburban areas (77.3%), regardless of BLL (Table 1). This is consistent with the distribution of Atlanta metro area residents, most of whom live in suburban areas.

In many respects, the three categories of BLLs (<2 µg/dL, 2 to <5 µg/dL, and ≥5 µg/dL) did not differ on area-level variables (Table 2). For example, the mean percentage of children aged 5 years or under living in any given census block was very similar, ranging 7.7–8.4%. Rates of whites and blacks across the different lead levels were in a tighter range than rates of whites and blacks at the individual level: White children in this study had rates (6.9%–13.9%) lower than blacks across all the lead levels (23.5–49.9%), and there was a much wider range of rates between blacks with low level (49.9%) to blacks with either sub-clinical (23.5%) or high (28.5%). The percentage of people with a GED or high school graduation within any given census block was 27.1–29.0%. Nevertheless, some differences were noted. There were somewhat fewer children with BLLs < 2 µg/dL and 2 to <5 µg/dL who lived in areas with a lower proportion of families with incomes below the poverty level (~23%), compared to children with BLLs ≥ 5 (26.6%). Furthermore, children with BLLs < 2 µg/dL and 2 to <5 µg/dL lived in areas with a lower proportion of houses built before 1980 (44.1% and 43.3%, respectively), compared to 52.6% in the group of children with BLLs ≥ 5 µg/dL. Similarly, compared to children with BLLs ≥ 5 µg/dL, children with the lower BLLs (2 to <5 µg/dL) lived in areas with a higher proportion of homeowners (45.1 ± 27.3 vs. 51.9 ± 27.2) and lower proportion of multi-unit structures (45.2 ± 31.3 vs. 38.2 ± 30.8).

### 3.3. Predictors of BLL 2 to <5 and BLL ≥ 5 µg/dL

To identify the predictors of a child having a sub-clinical BLL (2 to <5 µg/dL) compared to having a BLL < 2 µg/dL, we employed multilevel regression analysis. In Table 3, results of all 4 models are shown with the effect size: Model 1 indicated variability in the odds of having a BLL 2 to <5 µg/dL across blockID as compared with having a BLL < 2 µg/dL (τ = 0.3, *p* < 0.001). As indicated by the ICC, 8.4% of the variability in the odds of a child having a BLL 2 to <5 µg/dL was due to variability among census tracts. Model 2 (τ = 0.26, *p* < 0.001) yielded a PCV of 13.3%, indicating the amount of variance in the likelihood of a child having a BLL 2 to <5 µg/dL was explained by individual-level factors. Random parts of Model 3 showed that the odds of a child having a sub-clinical BLL remained statistically associated with unobserved area-level variability (τ = 0.29, *p* < 0.001), with 8.1% of the variance among the blockIDs due to area-level factors. Model 3 PCV indicated that 3.3% of the variance in the likelihood of having a BLL 2 to <5 µg/dL was explained by the specific area-level factors entered in the model. The variance at the area-level in Model 4 remained significant (τ = 0.24, *p* < 0.001) even after controlling for both individual and area-level factors. The ICC in Model 4 indicated that only 6.8% of the variation among the clusters was due to area-level factors. This decrease from 8.4% in Model 1 indicates inclusion of area-level variables was important for obtaining a better explanatory model. As shown in the PCV of Model 4, 20.0% of the variance in the odds of sub-clinical BLL across blockID was due to simultaneous effects of both individual and area-level factors found in Model 4. (Table 3). According to the AIC, Model 4 provided the best fit to the data. In this model, younger age, “other” race, non-Medicaid coverage, and sub-urban residence were the individual-level factors, while the proportion of renters and proportion of whites living within a 1.5-mile radius of a child’s residence were the area-level factors associated with a lower likelihood of a BLL 2 to <5 µg/dL compared to a BLL < 2 µg/dL. Conversely, the individual-level factors of being white or being of unknown race and rural residence were associated with a higher likelihood of a BLL 2 to <5 µg/dL.

We also aimed to identify the individual and area-level factors associated with the likelihood of a child having a sub-clinical BLL (2 to <5 µg/dL) compared to having a clinical BLL (≥5 µg/dL) using multilevel regression analysis. In Table 4, Model 1 (null model) showed variability in the odds of sub-clinical BLL across census blocks (τ = 0.79, *p* < 0.001). As indicated by the ICC across the four models, 16.1–19.4% of the variability in the odds of a child having a sub-clinical BLL was due to area-level factors. The variation in Model 2 remained significantly associated (τ = 0.72, *p* < 0.001), with 18.0% of the variance among observations being attributed to individual-level factors. Similarly, the PCV indicated that 8.8% of the variance in the likelihood of having a sub-clinical BLL was explained by the area-level factors. Random parts of Model 3 showed that the unobserved area-level variability in the odds of a child having a sub-clinical BLL remained (τ = 0.70, *p* < 0.001), with 17.5% of the variance among blockIDs being due to area-level factors. Similarly, the PCV indicated that 11.4% of the variance was explained by area-level factors. Finally, as shown in the PCV, 20.3% of the variance in the odds of having a sub-clinical BLL was due to simultaneous effects of both individual and area-level factors found in Model 4 (Table 4). According to the AIC, Model 4 provided the best fit to the data. In this final model, ‘other race’ and non-Medicaid status were the individual-level characteristics associated with lower likelihood of BLL 2 to <5 µg/dL over BLL ≥ 5 µg/dL, compared to younger age and unknown race which were associated with a higher likelihood of BLL 2 to <5 µg/dL. Among area-level characteristics, a higher proportion of pre-1980 housing stock in a given area and higher proportion of renters were related to lower odds of a sub-clinical BLL. Areas with higher proportion of GED/High school diploma were associated with higher odds of a sub-clinical BLL compared to BLL ≥ 5 µg/dL.

## 4. Discussion

Sub-clinical pediatric lead exposure is a persistent, but largely undocumented and therefore “silent” public health problem that has an associated impact on children’s health. BLLs of 2 µg/dL and above have been shown to negatively impact children’s neurodevelopment, resulting in adverse effects on cognitive function and behavior well into adulthood [14,26,27]. Most children in our study—those who had a venous BLL test—were Medicaid recipients and therefore at increased risk for elevated BLLs. In this group (*n* = 93,162) from the 20-county metro area, we observed a high representation of sub-clinical cases, 56.6%, compared to 2.3% of children with clinical BLLs. Given the current actionable BLL of 5 µg/dL, our findings indicate that a very small percentage of children will receive public health interventions, while the majority of children with significantly detectable levels of lead (2 to <5 µg/dL) remain under the radar to receive any such services. 

Our analysis of BLLs from 93,162 children aged 0–6 years old (0–72 months) in the 20-county metropolitan area of Atlanta, Georgia, from 2010 to 2018 revealed several insights. First, the Hotspot analyses showed that, between 2010 and 2018, inner-city and urbanized areas within the 20-county Atlanta metropolitan area made progress in reducing the number of children aged 0–72 months with BLLs ≥ 5 µg/dL. However, there are still largely concentrated areas of sub-clinical hotspots in the city-central area in addition to notable sub-clinical hotspots in the distal counties of the study area that have persisted over this 9-year time span. Second, the current screening guidelines appear to result in screening gaps for children aged 25–72 months, who comprised roughly 45% of total cases included in this study and who reflect younger aged children with both sub-clinical and elevated BLLs. Third, while black children did not appear to be more likely than whites to have elevated BLLs > 5 µg/dL, our findings in the ‘best fit’ model suggest racial differences were not a major driver in explaining different proportions of sub-clinical BLLs compared to elevated BLLs seen in certain geographic areas. Urban areas and the placed-based factors associated with living in urban locales may be more explanatory of the differences in the proportion of sub-clinical BLLs compared to elevated. Fourth, we identified salient predictors that distinguish children with BLLs 2 to <5 µg/dL from children with lower or higher BLLs (having insurance coverage other than Medicaid, the proportion of renters in each geographical area, and proportion of people with a GED/high school diploma). Some of these have been described previously as predictors of elevated BLLs (type of health insurance coverage, education level) [26,27]. These individual and area-level factors could be utilized to identify place-based risks associated with sub-clinical lead exposure and therefore, serve as the basis for developing targeted placed-based lead screening in at-risk children who might not otherwise be identified. We observed that nearly half of all the children with either significant sub-clinical BLLs or elevated BLLs were in the age range who would not have received routine screening based on current guidelines. This observation coupled with the placed-based risks identified with sub-clinical BLLs emphasizes the importance that screening should be guided by socio environmental conditions. Geographic information systems (GIS) technology has facilitated health disparities research and enabled mapping and colocalization of public health surveillance data with social determinants of health [28,29,30]. In 2012, the GDPH identified 14 counties with children at high risk for lead exposure [25]. Those county-level maps covered large, heterogeneous areas with broad-ranging risk factors that often change between neighborhoods. We show that in heterogeneous communities where gentrification and generational poverty exist in proximity, as in the metro Atlanta area, some of these risk factors persist, namely, health insurance coverage (often used as a proxy for individual-level SES), the proportion of renters, and proportion of people with GED/high school diploma (both of which could indicate area-level SES). While we did not observe an association of sub clinical BLLs with areas identified as higher risk for crime or higher proportion of young children, these factors have been identified as potential risks by others [31]. Sub-clinical BLLs persisted in specific areas of Atlanta’s urban center and the overall distribution pattern was similar across both younger age and older age groups; neighborhoods with a higher proportion of blacks did not significantly impact the overall rates of sub-clinical compared to elevated BLLs nor those with non-significant levels of lead. Moreover, over the 9 years, the general spatial distribution remained similar between 2010 and 2018 for sub-clinical BLLs. In contrast, spatial distribution of elevated BLLs across the age groups was different—our research did not uncover explanatory reasons why a larger spatial area was involved for older children with elevated BLLs. We also uncovered that the size of the spatial area where children with elevated BLLs resided differed dramatically in 2018 compared to 2010—regardless of age group, the area of involvement was smaller overall and clear hotspot regions persisted. This is an important finding especially as resources for public and primary care interventions are limited and targeting interventions may be more cost effective. In 2018 compared to 2010, we saw a relatively expansive area of involvement with older aged children with sub clinical BLLs compared to younger children, we did not find this to be the case with elevated. Children under 2 years with elevated BLLs seemed to reside over a larger area compared to older aged children during 2018, compared to 2010. Additional studies are needed to uncover how housing conditions, gentrification of communities, and other place-based risks which may account for this spatial pattern. Our spatial analyses further emphasize the importance of elucidating risks which are driven by geographic conditions rooted in socioenvironmental conditions faced by children.

The fact that hotspots of sub-clinical BLLs appear in suburban areas further indicates that lead exposure is not solely an urban issue, and closer attention should be paid to lead screening among children living in suburban areas of Atlanta and other large metropolitan areas. To identify at-risk children within these diverse areas, we may need to employ a more targeted approach to prevention activities, particularly in children >2 years. Pre-K and school-based interventions may be an example of how to leverage educational institutions to deploy prevention measures. 

Older children are not routinely tested in Georgia and, thus, sub-clinical BLLs in this population have the potential to become chronic. More children in our sample were younger (54.7%), likely reflecting practice guidelines under the Centers for Medicare and Medicaid Services (CMS) Medicaid program to test children at 12 and 24 months during well-child visits. From 2008 to 2014, approximately 30% of Medicaid-sponsored children were not tested by their second birthday [32,33]. Another study found that as few as 12% of children below 72 months were tested [34]. Our analysis and mapping of hotspots indicate that older children continue to be exposed to lead and at risk of having BLLs of 2 µg/dL or higher. These findings support the CMS report of widespread failure to meet current screening guidelines across the U.S. and suggests a need for increased screening and testing of older children [20]. This is crucial to lead prevention efforts in this vulnerable population and addresses a large screening gap that occurs during a highly neurodevelopmental phase in childhood. This screening gap is exacerbated by the lack of knowledge about the risk of sub-clinical BLLs in these age groups [35]. There is a misconception that risk assessment alone, determined through a questionnaire, is adequate to identify at-risk children [21]. The American Academy of Pediatrics’ recommendation is to test high-risk, asymptomatic children if they are 12 to 24 months old, and are living in neighborhoods where more than 25% of the houses were built before 1960 [36]. Our findings suggest that additional neighborhood-level information (residence in a suburban and urban area, % renters and % of people in an area with GED/high school diploma) would help expand the definition of at-risk areas. Other factors, like residential distance from major roadways may increase a child’s risk, given that soil in areas with historically heavy traffic has been linked to high lead levels; we did not explore this as a variable but these factors along with other place-based variables may need to be considered in future studies. Importantly, current guidelines do not recommend lead screening among privately insured children in the absence of suspected lead exposure during a wellness visit. Moreover, PeachCare and Medicaid do not provide reimbursement for testing which occurs outside of practicing guidelines, thereby disincentivizing screening in the absence of physician suspicion, or parental concern. In essence, our study demonstrates the high proportion of children who would have been ‘missed’ for either sub clinical or elevated lead levels under current screening policies.

Past studies evidenced that non-Hispanic Black children experienced higher mean BLLs compared to Mexican American and non-Hispanic White peer groups [26]. Due to the role of multiple factors around lead exposure distribution, such as place of residence, family, area poverty, and parental and area-level educational attainment, we cannot measure race as an isolated factor. However, many studies have shown that race makes a significant contribution to identifying high-risk children in developing lead prevention programs. In our study, race of the study children did not seem to be a major driver for risk of sub-clinical BLL compared to either low or elevated BLL; in our ‘best fit’ multi-level adjusted model, ‘other’ or ‘unknown’ race were more likely than either black or white children to be associated with sub-clinical BLLs over elevated BLLs. Additional studies are needed to delineate what factors are associated at both the individual and area levels and how they are related to the neighborhood or community from which children reside. As previously mentioned, gentrification and high heterogeneity of SES in the Atlanta area may explain some of the area-level factors which contributed to the distribution of sub-clinical BLLs compared to the low and elevated BLLs [37]. The “other race” designation included American Indian or Alaska Native, Asian or Pacific Islander, and Multiracial individuals. Because these groups are typically under-represented in population-based studies, it is difficult to draw firm conclusions about their risk of exposure in the 20-county metropolitan Atlanta area. Nevertheless, mean BLLs and the prevalence of BLL ≥ 5 µg/dL for these groups are lower than for other racial/ethnic groups participating in the NHANES, consistent with our findings [38].

In the current study, we observed a higher number of cases reported to the Georgia Healthy Homes and Lead Poisoning Prevention surveillance program from suburban areas, especially in areas where the houses were built before 1980. It is important to remark that large, densely populated counties like Fulton, DeKalb, Cobb, Clayton, and Gwinnett are primarily considered Urban-Suburban counties. Thus, the majority of all BLLs reported (81.6%) during this study period (2010 to 2018) were from Suburban areas, while only 18.1% were reported in Urban areas, based on self-reported residential zip code. Those counties also had a slight reduction in the total number of BLL reported for the study period. However, in more remotely located counties like Coweta, Carrol, Lamar, Meriwether, Newton, Talbot, and Upson, there was a predominance of sub-clinical BLLs within the observed increase in the total number of blood lead reported cases. The increased BLL reports could be due to the implementation of more robust surveillance programs in these areas, increases in parental awareness of the risk of exposure, or the movement of industrial settings from urban to rural areas. According to the CDC, there is a nationwide decline in the reported cases of elevated BLLs at the state levels (for Medicaid access children and in the general population). A decrease in the number of elevated or clinical BLLs does not necessarily correlate with a lower risk of sub-clinical lead exposure in children over time [20].

Our findings should be interpreted in light of study limitations and strengths. An important limitation is that the study population is a nonrandom sample, and thus, we cannot make generalized assumptions about all Georgia children. The dataset includes BLLs from the GDPH HHLPP data management platform which includes high-risk children targeted for screening by health care providers. Currently, these programs are not universal in their screening and reporting, but generally follow the CMS recommendations of testing Medicaid-sponsored children at 12- and 24-month well-child visits. Furthermore, the study sample consisted of different children at each time point, not allowing us to follow the individual children over time to assess the evolution in their BLLs as they aged. On the other hand, by drawing yearly samples from a similar baseline population over 9 years, we were able to document secular trends in the proportion of children with different levels of BLL between 2010 and 2018. It is also important to acknowledge that using the child’s reported residential address might not coincide with the child’s actual residence, especially for children of divorced or unmarried parents who split their time between multiple households. Similarly, the dataset had missing data on self-reported race, significantly contributing to the proportion of individuals with ‘unknown’ race, a persistent issue in utilizing laboratory-based surveillance datasets [39]. Lastly, the distribution of reported cases between counties was disproportional, resulting in some of the more rural counties having less representation within the study sample dataset. Fewer reported cases in those counties may reflect a low number of cases within those populations; it may also suggest that the Georgia Department of Public Health’s Healthy Homes and Lead Prevention Program (GDPH HHLPPP) surveillance program did not actively support/receive accurate reporting from those distal counties. On the other hand, the strengths of the study include: (1) the large sample size, (2) the focus on venous BLL results, (3) the comparison of likelihood of having BLL 2 to <5 µg/dL to both lower (<2 µg/dL) and higher (≥5 µg/dL) levels, (4) the testing of both individual and area-level predictors, (5) multi-level modeling that included both types of predictors in a single model, and (6) hotspot analysis that complemented the regression modeling.

## 5. Conclusions

Current lead testing guidelines miss at-risk children and contribute to gaps in access to reliable information about the prevalence of sub-clinical BLLs in children aged 0–72 months. Furthermore, there is no policy to require routine screening beyond the age of 24 months. In short, there is no way to protect children with sub-clinical BLLs from long-term intellectual and behavioral (neurocognitive) impairment associated with chronic lead exposure. Our study on the prevalence of sub-clinical BLLs in the Atlanta metro area suggests a “silent epidemic” that could be averted through better identification of at-risk areas and population groups, more universal lead testing even beyond the age of 2 years, and a lowering of the actionable BLL to 2 µg/dL.

## Figures and Tables

**Figure 1 ijerph-18-05163-f001:**
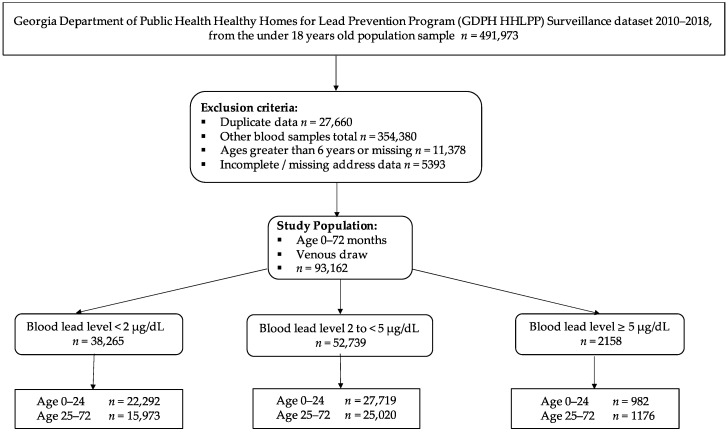
Study Schema for Georgia Department of Public Health Lead Prevention Program Surveillance Dataset from 2010 to 2018 for children aged 0–6 years old (0–72 months).

**Figure 2 ijerph-18-05163-f002:**
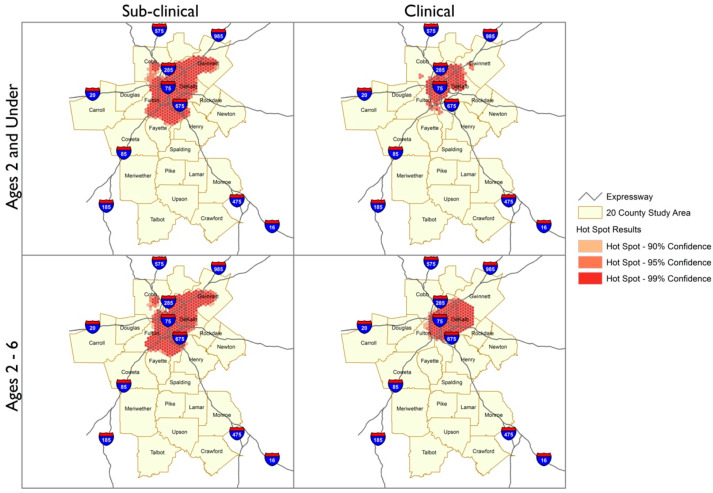
Comparison of Hotspot Analysis Distribution Patterns between Sub-clinical and Clinical BLLs for the year 2010. Hotspot analysis of the 90–99% confidence regions of hotspots for BLLs 2 to <5 µg/dL (Sub-clinical) and BLLs ≥ 5 µg/dL (Clinical) for 20-county metro Atlanta area in 2010. BLLs 2 to <5 µg/dL and BLLs ≥ 5 µg/dL are shown as red pattern dots with the darkest shaded regions reflecting the 99% confidence area.

**Figure 3 ijerph-18-05163-f003:**
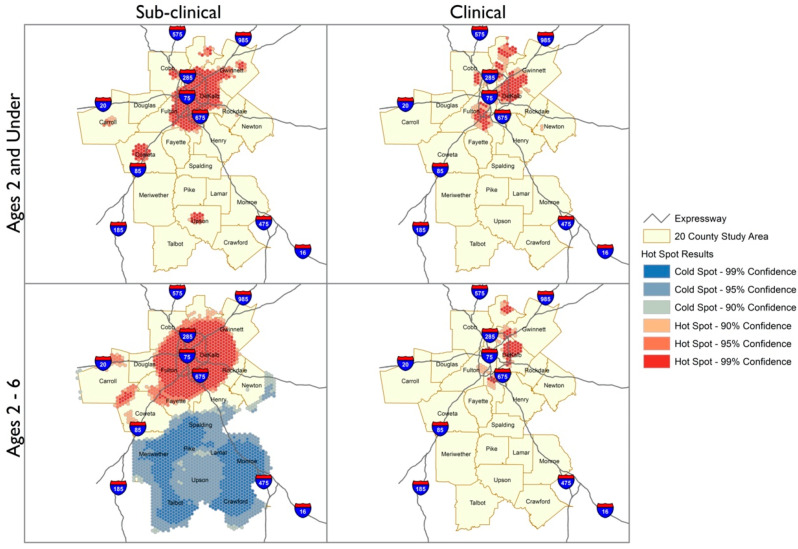
Comparison of Hotspot Analysis Distribution Patterns between Sub-clinical and Clinical BLLs for the year 2018. Hotspot analysis of the 90–99% confidence regions of hotspots for BLLs 2 to <5 µg/dL (Sub-clinical) and BLLs ≥ 5 µg/dL (Clinical) for 20-county metro Atlanta area in 2018. BLLs 2 to <5 µg/dL and BLLs ≥ 5 µg/dL are shown as red pattern dots with the darkest shaded regions reflecting the 99% confidence area.

**Table 1 ijerph-18-05163-t001:** Individual-level demographic characteristics of 93,162 children aged 0–72 months living in 20 counties in or near the Atlanta Metropolitan Statistics area who had a valid venous blood lead test in the years 2010–2018, by BLL category.

Individual-Level Variables	Total	BLLs < 2 µg/dL (*n* = 38,265) 41.1%	BLLs 2 to <5 µg/dL (*n* = 52,739) 56.6%	BLLs ≥ 5 µg/dL (*n* = 2158) 2.3%
		Frequency (%)	Frequency (%)	Frequency (%)
Age				
0–24 months	50,993(54.7)	22,292(58.3)	27,719(52.6)	982(45.5)
25–72 months	42,169(45.3)	15,973(41.7)	25,020(47.4)	1176(54.5)
Gender				
Female	45,268(48.7)	18,665(48.8)	25,577(48.6)	1026(47.7)
Male	47,782(51.4)	19,564(51.2)	27,092(51.4)	1126(52.3)
Race				
Black	31,731(34.6)	19,049(49.9)	12,073(23.5)	609(28.5)
White	9049(9.9)	5308(13.9)	3548(6.9)	193(9.0)
Other	10,248(11.2)	6302(16.5)	3542(6.9)	404(18.9)
Unknown	40,669(44.3)	7518(19.7)	32,222(62.7)	929(43.5)
Black	31,731(34.6)	19,049(49.9)	12,073(23.5)	609(28.5)
White	9049(9.9)	5308(13.9)	3548(6.9)	193(9.0)
Other	10,248(11.2)	6302(16.5)	3542(6.9)	404(18.9)
Unknown	40,669(44.3)	7518(19.7)	32,222(62.7)	929(43.5)
Medicaid status				
No	24,472(26.3)	10,629(27.8)	13,150(24.9)	693(32.1)
Yes	68,690(73.7)	27,636(72.2)	39,589(75.1)	1465(67.9)
Place of residence status				
Rural	334(0.4)	132(0.4)	191(0.4)	11(0.5)
Sub-urban	71,877(77.3)	30,169(78.9)	39,957(75.9)	1751(81.6)
Urban	20,835(22.4)	7941(20.8)	12,509(23.8)	385(17.9)

**Table 2 ijerph-18-05163-t002:** Area-level characteristics for the household location of 93,162 children aged 0–72 months living in 20 counties in or near the Atlanta Metropolitan Statistics area who had a valid venous blood lead test in the years 2010–2018, by BLL category.

Area-Level Variables	BLLs < 2 µg/dL	BLLs 2 to <5 µg/dL	BLLs ≥ 5 µg/dL
	Mean ± SD	Mean ± SD	Mean ± SD
Block-level			
Number of blocks	2075	2091	876
Number of individuals in the block (median, range)	11(1–177)	15(1–513)	1(1–118)
Ages of children in the block			
Under 5 years old (%)	7.7 ± 4.7	7.8 ± 4.6	8.4 ± 4.8
Racial composition			
White (%)	33.2 ± 27.7	31.6 ± 26.3	29.4 ± 26.3
Black (%)	55.3 ± 32.2	55.3 ± 31.4	50.8 ± 30.4
Poverty Status			
Income-Below Poverty (%)	22.9 ± 15.9	23.1 ± 15.8	26.6 ± 17.4
Educational Attainment (adults 25+ years)			
No High School Diploma (%)	16.6 ± 12.5	17.3 ± 12.8	21.1 ± 14.8
High School Diploma or equivalent (%)	29.0 ± 11.1	28.9 ± 10.6	27.1 ± 11.2
College Degree (%)	55.6 ± 17.0	55.1 ± 16.3	53.6 ± 17.3
Housing			
House Built before 1980 (%)	44.1 ± 27.7	43.3 ± 27.8	52.6 ± 28.2
Total Housing Units/100	10.4 ± 6.0	10.3 ± 6.1	9.2 ± 5.3
2010 Homeowner (%)	51.7 ± 27.2	51.9 ± 27.2	45.1 ± 27.3
2010 Renter (%)	48.3 ± 27.2	48.1 ± 27.2	54.9 ± 27.3
Single Family Home (%)	61.2 ± 31.1	61.8 ± 30.8	54.7 ± 31.4
Multi-unit homes (%)	38.8 ± 31.0	38.2 ± 30.8	45.2 ± 31.3
Household single female (%)	22.6 ± 11.1	22.8 ± 10.5	22.1 ± 9.7
Crime Status			
2019 Total Crime Index	173.1 ± 86.2	171.7 ± 87.6	164.7 ± 79.6

**Table 3 ijerph-18-05163-t003:** Individual and area-level predictors of the likelihood of having a BLL 2 to <5 µg/dL, compared to BLL < 2 µg/dL in 93,162 children aged 0–72 months living in 20 counties in or near the Atlanta Metropolitan Statistics area who had a valid venous blood lead test in the years 2010–2018.

Individual-and Area-Level Characteristics		AOR (95% CI)			
		Model 1	Model 2	Model 3	Model 4
Fixed effects					
Age	0–2	–	0.68(0.65–0.70)	–	0.68(0.66–0.70)
	2–6	–	1	–	1
Gender	Female	–	0.99(0.96–1.03)	–	0.99(0.96–1.03)
	Male	–	1	–	1
Race	Black	–	1	–	1
	White	–	1.20(1.13–1.28)	–	1.31(1.22–1.39)
	Other	–	0.83(0.79–0.88)	–	0.85(0.80–0.90)
	Unknown	–	9.78(9.36–10.23)	–	10.05(9.61–10.51)
Medicaid status	No	–	0.42(0.40–0.44)	–	0.42(0.40–0.44)
	Yes	–	1	–	1
Area	Suburban	–	0.91(0.85–0.98)	–	0.91(0.85–0.98)
	Rural	–	1.25(0.89–1.75)	–	1.48(1.06–2.07)
	Urban	–	1	–	1
Age under 5 (%)		–	–	1.06(0.99–1.14)	1.03(0.96–1.11)
Houses Built pre-1980 (%)		–	–	0.98(0.97–1.00)	1.01(0.99–1.02)
2019 Total Crime Index		–	–	1.00(0.99–1.00)	1.00(0.99–1.00)
2010 Renter (%)		–	–	0.98(0.97–0.99)	0.98(0.96–0.99)
GED/High School Diploma (%)		–	–	1.02(0.99–1.04)	1.02(0.99–1.04)
White (%)		–	–	0.97(0.96–0.98)	0.93(0.92–0.95)
Block Random effects					
Block variance (SE)		0.30(0.02)	0.26(0.02)	0.29(0.01)	0.24(0.01)
ICC (%)		8.4	7.3	8.1	6.8
PCV (%)		Ref	13.3	3.3	20.0
Model fit statistics					
Log-likelihood		120,633.5	81,475.32	120,574.7	81,349.43
AIC		120,637.5	81,495.32	120,590.7	81,381.43

**Table 4 ijerph-18-05163-t004:** Individual and area-level predictors of the likelihood of having a BLL 2 to <5 µg/dL compared to BLL ≥ 5 µg/dL, in 93,162 children aged 0–72 months living in 20 counties in or near the Atlanta Metropolitan Statistics area who had a valid venous blood lead test in the years 2010–2018.

Individual-and Area-Level Characteristics			AOR (95% CI)			
			Model 1	Model 2	Model 3	Model 4
Fixed effects						
Age	0–2	–		1.13(1.03–1.25)	–	1.13(1.02–1.25)
	2–6	–		1	–	1
Gender	Female	–		0.98(0.89–1.08)	–	0.97(0.88–1.10)
	Male	–		1	–	1
Race	Black	–		1	–	1
	White	–		0.85(0.70–1.03)	–	0.90(0.73–1.10)
	Other	–		0.80(0.68–0.94)	–	0.83(0.71–0.98)
	Unknown	–		1.77(1.57–2.00)	–	1.83(1.62–2.06)
Medicaid status	No	–		0.64(0.58–0.72)	–	0.66(0.59–0.73)
	Yes	–		1	–	1
Area	Suburban	–		0.79(0.66–0.95)	–	0.92(0.76–1.11)
	Rural	–		0.78(0.32–1.94)	–	0.78(0.32–1.85)
	Urban	–		1	–	1
Age under 5 (%)		–		–	1.08(0.91–1.27)	1.12(0.95–1.33)
Houses Built pre-1980 (%)		–		–	0.92(0.90–0.95)	0.92(0.90–0.95)
2019 Total Crime Index		–		–	1.01(1.00–1.02)	1.01(1.00–1.02)
2010 Renter (%)		–		–	0.96(0.93–0.99)	0.96(0.93–0.99)
GED/High School Diploma (%)		–		–	1.08(1.02–1.15)	1.10(1.02–1.17)
White (%)		–		–	0.98(0.95–1.01)	0.99(0.96–1.03)
Block Random effects						
Block variance (SE)		0.79(0.07)		0.72(0.08)	0.70(0.07)	0.63(0.07)
ICC (%)		19.4		18.0	17.5	16.1
PCV (%)		Ref		8.8	11.4	20.3
Model fit statistics						
Log-likelihood		17,029.17		14,005.37	16,977.06	13,966.2
AIC		17,033.17		14,025.37	16,993.06	13,998.2

## Data Availability

Data sharing not applicable. No new data were created or analyzed in this study.

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
