# Peer review of "Increased Risk of Sub-Clinical Blood Lead Levels in the 20-County Metro Atlanta, Georgia Area—A Laboratory Surveillance-Based Study"

_ijerph, 2021, doi:10.3390/ijerph18105163_

Round 1

Reviewer 1 Report

Very interesting topic, BLL and subclinical evaluation. Is there any study, regarding geological Lead determination in soil of Gerrgia ? 

Lead is ubiquos metal, present in food, water... Is there any residual evaluation, in that matrixes ?

Author Response

Upon performing a literature search, there are no articles which cite soil lead levels in the Atlanta metro area, although Deocampo et al. did a report on the extent of lead pollution in road dust in two Atlanta neighborhoods (2012). I agree that this is a relevant piece of information to have, and we think it should be considered in future studies.

Nevertheless, the focus of this paper was to understand how well demographic variables, both at the individual and census-tract level, perform as predictors of BLLs <5 and =>5 ug/dL. The reason this is of interest is that these variables, unlike results of geological surveys, are most readily available to public health officials.

Reviewer 2 Report

General comment: The manuscript brings new and useful data about the distribution of lower to sub-clinical blood lead levels (BLLs) (0 µg/dL to < 5 µg/dL) in children ages 0–72 months and their associations with sociodemographic and area-level variables in the 20-county Metro Atlanta, Georgia. The results are interesting, and the authors claimed that about half of the children in the study are at risk for adverse outcomes associated with sub-clinical BLLs. The present study seems interesting and potentially could contribute to the research field and will help to policy intervention. However, direct adverse outcomes or health consequences due to sub-clinical lead exposure in children are missing in this study.

Specific comments:

Title

The title is informative and relevant to the major findings.

Abstract

In the abstract, the aim of the study clearly mentioned. Major results/findings and recommendations are not adequately presented. 

Introduction

The research question/gap is clearly outlined. The introduction may improve further.

Materials and Method

Well described.

 Results

The results are not well structured. Subsections 3.1 Spatial analyses and 3.2 Data Analyses maybe not appropriate in the results section.

Discussion

The discussion section could be more focused.

Conclusion

Conclusion properly answered the aims of the study and supported by the results. Major limitation and opportunities to inform future research are addressed.  

Overall comments

There are some typos, for example, Line 46 (10ug/dL), Line 99 (n 11,378) = missing.

Author Response

We thank the reviewer for these comments and recognition of the value our analysis brings to the issue of lead exposure in young children. We agree that lack of information on health outcomes is a drawback. Given the nature of the dataset, obtained from the Georgia Department of Public Health Healthy Homes for Lead Prevention Program, where health data are not collected, we are unable to make direct connections between exposure and health of the study children. We did, however, point to other studies that have linked blood lead levels <5 ug/dL with cognitive abilities and behavior in children (see introduction).

Specific comments:

  1. Title: The title is informative and relevant to the major findings.

Thank you, no changes have been made.

  1. Abstract: In the abstract, the aim of the study clearly mentioned. Major results/findings and recommendations are not adequately presented. 

Thank you, no changes have been made.

  1. Introduction: The research question/gap is clearly outlined. The introduction may improve further.

Thank you, we have reviewed our introduction and revised to clarify our points throughout.

  1. Materials and Method: Well-described.

Thank you, we’ve reviewed our methods section and made small revisions to improve clarity.

  1. The results are not well structured. Subsections 3.1 Spatial analyses and 3.2 Data Analyses maybe not appropriate in the results section.

Thank you for these comments. We have elected to retain spatial analyses in the results section as they directly address the change in the prevalence of BLL 2-4.9 ug/dL and =>5 ug/dL over the 9-year study period. We agree, however, that the section 3.2, as named currently appears not to belong in the results section. We have renamed section 3.2 as “Participant Characteristics”. We have also added section 3.3 “Predictors of BLL 2 to <5 ug/dL and => 5 ug/dL”

  1. Discussion: The discussion section could be more focused.

Thank you, we have gone through the discussion section and revised for succinctness.

  1. Conclusion: Conclusion properly answered the aims of the study and supported by the results. Major limitation and opportunities to inform future research are addressed.

Thank you, no changes have been made.  

  1. Overall comments: There are some typos, for example, Line 46 (10ug/dL), Line 99 (n 11,378) = missing.

Thank you, we have revised accordingly.

Reviewer 3 Report

The exposures to environmental toxic metals of hydro-geological origin such as arsenic, lead, cadmium, mercury and copper etc. having potential deleterious effects on human health and pose the risk of deadly disease such as cancers in the bladder, kidney, liver, lung, skin and cardiovascular disease etc. Several recent studies suggest that the human exposures with these toxic metals rises as serious global public health problems. In present scenario, the topic of the manuscript is very important and is need of the current situation. However, the manuscript contains few lacunas which need to be justified. Therefore, my decision is "Minor Revision".

Following are comments for consideration by authors:

  1. Authors need to mention in the manuscript the full name of GDPH HHLPPP and further abbreviation of it can be used.
  2. Results presented in Table 1, the blood lead levels (BLLs) is high in African-American population. What are the possible reasons for it? Need to explain in discussion section.
  3. The discussion of the manuscript is appropriate but authors need to discuss it more with previous related clinical findings.

Author Response

Following are comments for consideration by authors:

  1. Authors need to mention in the manuscript the full name of GDPH HHLPPP and further abbreviation of it can be used.

Thank you, we have done this on lines 98-9.

  1. Results presented in Table 1, the blood lead levels (BLLs) is high in African-American population. What are the possible reasons for it? Need to explain in discussion section.

Thank you for this comment. This table does not present actual BLLs in different population groups. Rather, it presents the prevalence of BLL <2, 2 to < 5 and =>5 ug/dL in different groups. It appears that a higher percentage of African American children is in the <2 ug/dL group compared to other race groups. On the other hand, more children with “unknown” race have BLLs in the range of 2 to <5 or >=5 ug/dL compared to African Americans or other races.

  1. The discussion of the manuscript is appropriate but authors need to discuss it more with previous related clinical findings.

Thank you for this suggestion. The purpose of our work was to understand if individual or census-tract predictors could be used to identify children with BLLs >2 ug/dL. We have discussed our findings with this goal in mind. Therefore, we have focused our discussion on the utility of our predictors and on the public health/programmatic implications of recognizing BLL >2 ug/dL as an important cut-point for public health action. We do discuss our findings with respect to clinical practice related to lead screening and testing.

Reviewer 4 Report

It was my pleasure to review the manuscript entitled ‘Increased Risk of Sub-clinical Blood Lead Levels in the 20-county Metro Atlanta, Georgia area – A Laboratory Surveillance-Based Study’, which examined possible contributing factors for children’s blood lead levels. The manuscript is well written, and topics presented is very important. I have minor comments to improve this manuscript.

  1. Units and punctuations should be consistent throughout the manuscript. For example, 10 µg/dL instead of 10ug/dL (line 46), n = instead of n (line 99), quotation marks, space before %, so on.
  2. Please provide method performance characteristics of lead measurements or any information about the credentials of laboratories that conducted the lead measurements.
  3. Authors use the term ‘significant’ or ‘significance’. Do these indicate ‘statistical significance’? As this study included a large number of subjects, p-values should become small. The small p-values do not indicate the scientific or public health significance of the results. Please refer to the ASA Statement on Statistical Significance and P-Values: https://amstat.tandfonline.com/doi/full/10.1080/00031305.2016.1154108.
  4. What does the asterisk mean in Tables 3 and 4?
  5. Please add some discussion about biological half-life of lead. For example, the discussion starting in line 374 needs consideration about the lead half-life since lead accumulates in body over the years.

Author Response

  1. Units and punctuations should be consistent throughout the manuscript. For example, 10 µg/dL instead of 10ug/dL (line 46), n = instead of n (line 99), quotation marks, space before %, so on.

Thank you, we have reviewed and corrected these details as appropriate.

  1. Please provide method performance characteristics of lead measurements or any information about the credentials of laboratories that conducted the lead measurements.

Thank you for this comment, unfortunately we do not have access to this data.

  1. Authors use the term ‘significant’ or ‘significance’. Do these indicate ‘statistical significance’? As this study included a large number of subjects, p-values should become small. The small p-values do not indicate the scientific or public health significance of the results. Please refer to the ASA Statement on Statistical Significance and P-Values: https://amstat.tandfonline.com/doi/full/10.1080/00031305.2016.1154108.

Thank you for this suggestion. We have revised our manuscript to refer to “statistical associations” rather than significance. See line 252.

  1. What does the asterisk mean in Tables 3 and 4?

Thank you for this question, the asterisks in tables 3 and 4 have been removed.

  1. Please add some discussion about biological half-life of lead. For example, the discussion starting in line 374 needs consideration about the lead half-life since lead accumulates in body over the years.

Thank you for this observation. The authors agree that We have included a statement about the accumulation of lead in bone over time in lines 46-47.